# SelfJudge: Faster Speculative Decoding via Self-Supervised Judge Verification

## Abstract

Speculative decoding accelerates LLM inference by verifying candidate tokens from a draft model against a larger target model. Recent "judge" decoding boosts this process by relaxing verification criteria by accepting draft tokens that may exhibit minor discrepancies from target model output, but existing methods are restricted by their reliance on human annotations or tasks with verifiable ground truths, limiting generalizability across diverse NLP tasks. We propose `SelfJudge`, which trains judge verifiers via self-supervision of the target model. Our method measures semantic preservation by assessing whether token-substituted responses preserve the meaning of original responses, enabling automatic verifier training across diverse NLP tasks. Our experiments show `SelfJudge` achieves superior inference-accuracy trade-offs than judge decoding baselines, offering a broadly applicable solution for faster LLM inference.

## 1 Introduction

Modern large language models (LLMs) have made significant strides in natural language processing, demonstrating competitive performance across a wide range of tasks (Radford et al., 2019; OpenAI et al., 2024). Empirical scaling laws establish a relationship between the number of parameters and model capability, as evidenced by models with hundreds of billions of parameters achieving state-of-the-art results on benchmarks (Kaplan et al., 2020; Grattafiori et al., 2024). However, the autoregressive generation process requires accessing all model parameters for each forward pass, creating a memory bandwidth bottleneck that impacts token generation latency. Furthermore, current trends toward more sophisticated LLM applications, such as multi-hop reasoning (Wei et al., 2022), tool integration (Patil et al., 2024), and reasoning capability (Yang et al., 2025; DeepMind, 2025), produce longer output sequences, amplifying the computational burden of inference.

One prominent approach to address inference latency is Speculative Decoding (SD), which achieves partial parallelization of the generation process (Leviathan et al., 2023; Chen et al., 2023). Standard SD operates by deploying a computationally efficient *draft* model to propose candidate token sequences, which are subsequently validated in parallel by the *target* model (the model of interest). The acceptance criterion for draft tokens relies on a probability-based alignment verification: draft tokens are accepted when their likelihood under the target model meets or exceeds their likelihood under the draft model. This probabilistic acceptance mechanism guarantees distributional equivalence with standalone target model generation, preserving output quality while achieving computational speedup (Li et al., 2024a; Sun et al., 2023; 2024; Hu et al., 2025; Zhou et al., 2024).

Recently, Bachmann et al. (2025) and Garipov et al. (2025) revealed that alignment-based verification approaches are overly conservative and lack contextual awareness. Specifically, when minor lexical variations occur between draft and target models without altering the contextual meaning, conventional SD rejects these draft tokens. The conservative behavior limits the speedup potential of SD even when output quality remains uncompromised. To address the limitation, Bachmann et al. (2025) proposed the judge decoding that relaxes strict token alignment by evaluating semantic compatibility through a learnable *judge verifier*, enabling greater inference acceleration.

However, judge verification approaches face substantial practical limitations in judge verifier training. Effective judge verifiers require token-level supervision data indicating acceptance decisions for individual tokens, a challenging annotation requirement. Recent approaches have attempted to address this through human annotation (Bachmann et al., 2025) or automated labeling based on

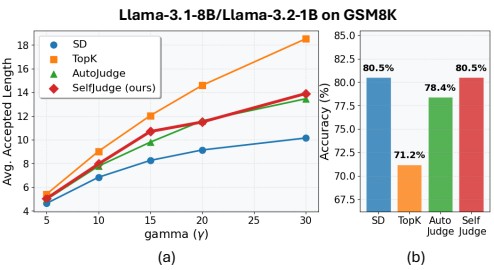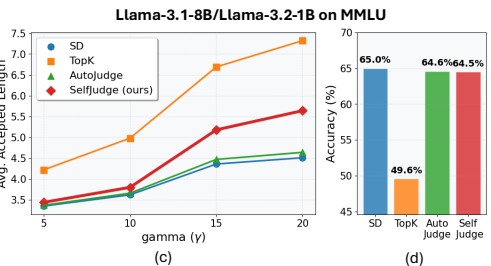

Figure 1: Inference efficiency and task performance comparison of SD methods on GSM8K (a,b) and MMLU (c,d). AutoJudge shows domain-specific limitations, performing well on mathematical reasoning but poorly on general knowledge tasks, while `SelfJudge` maintains consistent performance across both domains. $\gamma$ represents the number of tokens generated by draft model per step.

task correctness (Garipov et al., 2025), but both strategies exhibit fundamental constraints. Human-annotated approaches introduce bias due to annotation subjectivity and divergence from target model decisions. In contrast, automated methods are constrained to tasks with verifiable ground truth, restricting applicability to specific domains such as mathematical reasoning and code generation. This domain constraint is demonstrated empirically in Figure 1, where automated approaches such as AutoJudge achieve substantial speedup on mathematical reasoning tasks (GSM8K) but exhibit significant performance degradation on general knowledge evaluation (MMLU).

In this paper, we propose `SelfJudge`, a speculative decoding framework that enables judge verification across general NLP tasks without requiring human supervision or verifiable ground truth. Our approach exploits self-supervision from the target model to generate verifier training data. We introduce a criterion for token acceptability based on semantic preservation, quantified as the likelihood difference between the original target response and token-substituted variants under the target model's distribution. Tokens whose replacement minimally affects the target model's response likelihood are labeled as acceptable for training; tokens that substantially alter likelihood are rejected. This semantic preservation criterion enables automatic supervision generation across diverse task domains, including open-ended question answering and text summarization. The generated data is used to train a lightweight verifier that performs context-aware token evaluation during inference, accepting draft proposals based on semantic compatibility rather than distribution alignment.

We evaluate `SelfJudge` on diverse NLP tasks and datasets, including GSM8K, MATH-500, MMLU, CNN/DailyMail and LiveCodeBench. We demonstrate that `SelfJudge` outperforms existing verification methods by achieving superior accuracy-efficiency trade-offs across domains. While prior judge verification methods like AutoJudge achieve +1.96 accepted tokens per verification cycle with significant task performance degradation (-2.7% accuracy), our approach delivers enhanced +2.06 accepted tokens per verification cycle with only -1.0% accuracy. The improvement demonstrates `SelfJudge`'s superior applicability to general-purpose NLP scenarios, maintaining task quality while enabling efficient speculative decoding across diverse domains.

In summary, we make the following contributions:

- We propose `SelfJudge`, a novel judge verification approach that enables generalization across diverse NLP tasks by accepting semantically coherent output tokens for faster LLM inference.

- We introduce an automatic training data generation method for the judge verifier that labels tokens using a semantic preservation score, measured by the target model's likelihood differences.

- We conduct comprehensive experiments across NLP tasks using the Llama-3 and Qwen-2.5 families, demonstrating that `SelfJudge` generally outperforms existing verification methods in terms of token generation speed and task performance.

## 2 RELATED WORKS

### 2.1 SPECULATIVE DECODING

SD is a partial parallelization approach that addresses the latency of autoregressive generation. The core idea is to use an efficient module to generate drafts, which are then verified in parallel by the target model. Existing SD methods can be categorized into parameterized and parameter-free approaches. First, several SD methods have used small language models from the same family to suggest drafts (Leviathan et al., 2023; Miao et al., 2024). To enhance the alignment between draft and target models, advanced approaches propose specialized drafting architectures that utilize parts in the target model (e.g., hidden representation, LM head) (Stern et al., 2018; Cai et al., 2024; Li et al., 2024a; 2025). Moreover, recent approaches tried to improve the distribution alignment through knowledge distillation, which improves the draft acceptance rate (Zhou et al., 2024; Liu et al., 2024). In contrast, parameter-free SD methods produce a draft using tokens from user prompts or generated history (Fu et al., 2024; Luo et al., 2025; Oliaro et al., 2025). While previous approaches have focused on improving the drafting phase to generate draft tokens that the target model is likely to accept, our work focuses on developing an improved verification method that allows the acceptance of these tokens without altering their semantic meaning. In this paper, our goal is to achieve speedup while maintaining task performance through our proposed verification approach.

### 2.2 JUDGE SPECULATIVE DECODING

Standard SD methods perform draft verification based on modified rejection sampling (Chen et al., 2023), which accepts tokens when the token probabilities from the target model meet or exceed those of the draft model. The advantage of using rejection sampling is that it preserves the distribution of the target model without degrading output quality. However, this alignment-based verification is overly conservative, which limits the speedup of SD when there are minor token discrepancies between draft and target models. To address this limitation, JudgeDecoding (Bachmann et al., 2025) have been proposed. JudgeDecoding utilizes a lightweight classifier as verifier, which examines the hidden representations of tokens. However, training such verifiers remains challenging as it requires labels indicating whether each token should be accepted or not. Hence, AutoJudge (Garipov et al., 2025) automates the data generation process by using verifiable ground truth in math and coding tasks. Specifically, AutoJudge adopts the answer-preservation approach, where it systematically tests token alternatives while maintaining correctness of math and coding problems. The method replaces individual tokens in a correct response and completes the modified response. If the final answer remains correct, it considers the replaced token as acceptable. Despite these advances, current judge verification methods lack generalization ability across diverse NLP tasks, showing performance improvements primarily on math and coding tasks. In this paper, we propose a judge verification approach that trains the verifier using semantic coherence from the original response, aiming to improve generalization across diverse NLP domains.

## 3 METHOD

In this section, we propose `SelfJudge`: a novel speculative decoding approach that enables judge verification across general NLP tasks without requiring human annotations or ground truth answers. The key innovation of `SelfJudge` lies in leveraging the semantic information inherent in the target model to automatically generate verifier training data.

As shown in Figure 2, `SelfJudge` introduces a self-data-generation paradigm where the target model assesses the semantic change of the original response with a token substituted. Specifically, we define and measure semantic preservation by comparing the likelihood in the target model's original response versus token-substituted variants. Tokens whose replacement does not significantly affect the model's overall response confidence are labeled as acceptable, enabling automatic generation of high-quality training data across diverse NLP domains. This semantic preservation-based approach fundamentally shifts the verification paradigm from exact token matching or task-specific correctness to general semantic preservation. By obtaining the semantic preservation in the perspective of the target model, our approach makes it applicable to any NLP tasks, including open-ended tasks (e.g., creative writing) where ground truth is unavailable.

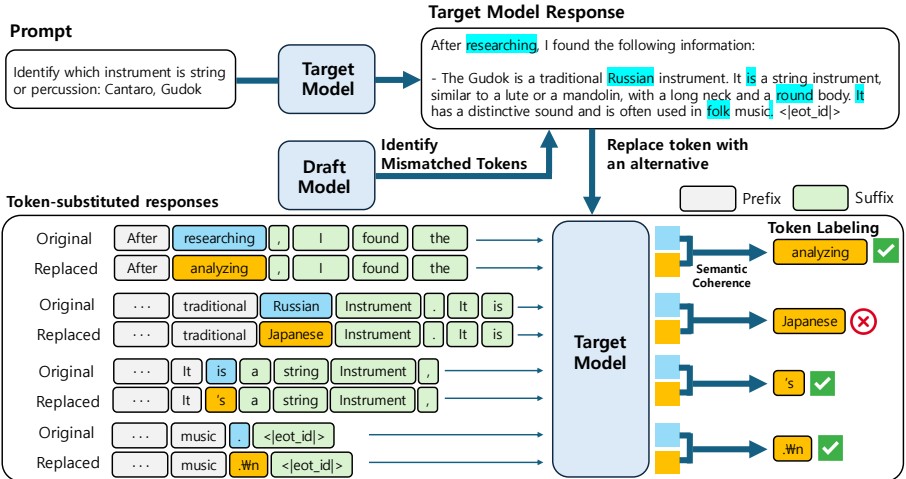

Figure 2: The training data generation process of SelfJudge for the verifier. Our approach compares the likelihood of the replaced response with the original response to measure the semantic preservation score. If semantic preservation score is higher than $\tau$, the replaced token is labeled as acceptable. After the token labeling process, we train the verifier that will be used during the inference phase for draft verification.

## 3.1 CONVENTIONAL SPECULATIVE DECODING

We begin by introducing conventional SD. The core idea of SD is to efficiently generate drafts and verify them in parallel. Specifically, we generate $\gamma$ candidate tokens using a small language model (Leviathan et al., 2023) or a specialized autoregressive model (Li et al., 2024a) to create drafts. When these generated candidate tokens are fed into the target model, we can verify whether the tokens are sufficiently aligned with the target model's output using the probability value of next token.

**Draft Generation.** Formally, let $\mathcal{M}_{\text{target}}$ be the large language model of interest, and $\mathcal{M}_{\text{draft}}$ be the efficient model with a smaller number of parameters compared to $\mathcal{M}_{\text{target}}$. In the drafting phase, given prefix $\mathbf{x}_{\leq i}$, we first generate $\gamma$ tokens using the efficient model autoregressively:

$$(d_{i+1}, q_{i+1}) = \mathcal{M}_{\text{draft}}(\mathbf{x}_{\leq i}), \quad \text{where } i = t, t+1, \ldots, t+\gamma - 1 \tag{1}$$

where $d_{i+1}$ represents the generated draft token and $q_{i+1}$ denotes its corresponding probability.

**Alignment-based Verification.** Subsequently, we feed the draft tokens into the target model to verify whether each token is valid from the target model's perspective:

$$(p_{t+1}, \ldots, p_{t+\gamma}, p_{t+\gamma+1}) = \mathcal{M}_{\text{target}}([\mathbf{x}_{\leq t}; d_{t+1}, ..., d_{t+\gamma}]) \tag{2}$$

where $p_t$ represents the target model's probability value for each draft token $d_t$. The last probability $p_{t+\gamma+1}$ corresponds to additionally sampled token using the target model.

Conventional SD typically employs rejection sampling to verify draft tokens by comparing probability distributions from both the draft and target models for each token, accepting tokens only when the target model probabilities are high or similar to the draft probabilities as follows:

$$\text{Accept } d_t \text{ if } u_t < r_t, \text{ otherwise reject } d_t \quad \text{where} \quad r_t = \min\left(1, \frac{p_t}{q_t}\right), \quad u_t \sim \text{Unif}(0, 1) \tag{3}$$

From the draft sequence, we accept tokens sequentially until the first rejection occurs according to Eqn.(3). It ensures that the accepted draft tokens maintain the target distribution. Beyond rejection sampling, other alignment-based methods exist, such as strict speculative sampling (He et al., 2024), which accepts draft tokens only when they match the target model's greedy predictions exactly.

## 3.2 JUDGE VERIFICATION

As alignment-based verification methods have strict criteria, they lack context-awareness. This strict criterion can be overly conservative, for instance, rejecting *"It's a strong instrument"* when the

target model produces *"It is a strong instrument"*, even though both expressions preserve the same semantic meaning in context. Such unnecessary rejections limit the potential speedup of SD.

To address this issue, JudgeDecoding (Bachmann et al., 2025) introduces judge verification to complement alignment-based methods by recovering semantically acceptable tokens that would otherwise be conservatively rejected. Given draft tokens $d_{t+1}, \ldots, d_{t+\gamma}$, we first obtain the hidden representations of the target model:

$$(h_{t+1}, \ldots, h_{t+\gamma}) = \mathcal{M}_{\text{target}}([\mathbf{x}_{\leq t}; d_{t+1}, \ldots, d_{t+\gamma}]) \tag{4}$$

The judge verifier then evaluates each token based on its hidden representation:

$$\text{Verifier}(h_t) > \theta \implies \text{Accept } d_t \tag{5}$$

where Verifier is a learned binary classifier (we use logistic regression for simplicity), and $\theta$ is a hyperparameter determined via holdout validation.

In practice, judge verification operates in parallel with alignment-based verification (e.g., rejection sampling). A token is accepted if *either* the judge verifier or the alignment-based method approves it, ensuring that semantically coherent tokens are not unnecessarily discarded while maintaining the distributional guarantees of alignment-based verification as a safety net.

### 3.3 GENERATE DATA FOR JUDGE VERIFIER

For efficient and accurate judge decoding, obtaining a high-quality judge verifier is paramount. Our core insight is to leverage the target model's own semantic understanding to train the verifier. By doing so, we increase the accepted tokens by allowing semantically valid tokens that may have a lower probability in the target model while maintaining response quality. Consequently, the judge verifier must identify which tokens preserve the original semantic meaning within the given context.

#### 3.3.1 SEMANTIC PRESERVATION-BASED TOKEN LABELING

To maintain response quality in judge verification, previous methods have relied on human annotators (Bachmann et al., 2025) or on whether the final answers are preserved when tokens are replaced (Garipov et al., 2025). However, human annotations suffer from subjectivity, and the answer-preservation-based method is limited to tasks with verifiable ground truths.

Rather than relying on external supervision, we utilize the target model's self-evaluation. Since the goal of SD is to preserve the target model's generation quality while improving latency, the target model itself is the best judge of semantic preservation. In our approach, the target model directly assesses the semantic impact of token replacements within its own responses. This enables automatic generation of high-quality training data across diverse NLP tasks without domain constraints.

**Step 1. Identifying Mismatched Tokens.** To improve the efficiency of SD with a verifier, we first choose which tokens to label, focusing on mismatched tokens where the target and draft models disagree. These tokens are candidates for relaxed verification, whereas matched tokens are already handled by the alignment-based verification phase of SD.

Formally, we generate response $y$ using the target model given an instruction. We then feed response $y$ to the draft model and obtain the draft model's predictions at each position. We identify mismatched positions $i$ where the target token $y_i$ differs from the draft model's top prediction $\arg\max P_{\text{draft}}(\cdot|y_{<i})$.

**Step 2: Computing Semantic Preservation Score.** When the draft model's token does not match the token output by the target model, we need to determine if the alternative token from the draft model, i.e., $\arg\max P_{\text{draft}}(\cdot|y_{<i})$), is a semantically acceptable substitute to the target model. To this end, we introduce a *semantic preservation score*. Our core insight is to consider the alternative token as acceptable if it preserves the semantic meaning of the target model's response. Therefore, our semantic preservation score quantifies how much of the semantic meaning of the target model's response is preserved when the target model's original token is replaced by the draft's alternative.

Formally, given the target response $y$ and the mismatched index $i$, we define the following *semantic preservation score* for token replacement $z_i$ as:

$$s(y, z_i) = \log P(z_i|y_{<i}, y_{>i}) - \log P(y_i|y_{<i}, y_{>i}) \tag{6}$$

where $z_i$ is the alternative token from the draft model and $y_{>i}$ represents the future token sequence. In practice, our semantic preservation score uses $N$ future tokens as opposed to using all future tokens. This score measures how semantically similar the replacement token is to the target model's established context. To illustrate this, see the example in Figure 2. The original response is "`After` **`researching,`** `I found the following information ...`" and the token replacement proposed by the draft model is **researching** $\rightarrow$ **analyzing**. Our semantic preservation score computes the likelihoods that **researching** and **analyzing** appear between the prefix "`After`" and the suffix "`, I found the following information`", respectively. When `analyzing` is semantically similar to `researching` in this bidirectional context, the target model assigns similar probabilities to the original and the alternative token, resulting in a *high* preservation score. If a semantically inappropriate token is given as an alternative token, the target model assigns a *low* preservation score. This infilling-style token scoring allows us to evaluate the semantic preservation of a token replacement with a richer context of the target model. This bidirectional scoring is used solely for generating verifier training labels; during inference, the trained verifier operates on forward-pass without access to future tokens.

As many modern LLMs have adopted autoregressive inference, directly computing the probabilities conditioned on the bidirectional context in Eqn.(6) is not possible. Therefore, we propose a method of transforming the bidirectional semantic difference into computationally tractable autoregressive operations as follows:

$$
\begin{aligned}
&\log P(z_i|y_{<i}, y_{>i}) - \log P(y_i|y_{<i}, y_{>i}) \\
&= \log P(z_i, y_{<i}, y_{i>i}) - \log P(y_i, y_{<i}, y_{i>i}) \\
&= \log \cancel{P(y_{<i})} P(z_i|y_{<i}) P(y_{>i}|y_{<i}, z_i) - \log \cancel{P(y_{<i})} P(y_i|y_{<i}) P(y_{i>}|y_{\leq i}) \\
&= \underbrace{\log P(z_i|y_{<i}) - \log P(y_i|y_{<i})}_{s_{\text{prefix}}} + \underbrace{\log P(y_{>i}|y_{<i}, z_i) - \log P(y_{>i}|y_{\leq i})}_{\text{suffix likelihood}}
\end{aligned}
\tag{7}
$$

$s_{\text{prefix}}$ represents the relative probability given the prefix, computing the difference between the original and replaced token probabilities given the same prefix. The second term represents future context consistency, which computes the likelihood difference of future tokens in the target model's response. Note that we obtain the probabilities conditioned on the replaced token by directly feeding the token-substituted response, i.e., $[y_{<i}, z_i, y_{>i}]$, into the target model.

**Step 3: Token Labeling.** Based on the semantic preservation score computed on the mismatched tokens, we finally obtain the label for each token. Specifically, a replaced token is labeled as acceptable if $s(y, z_i) > \tau$. Otherwise, the replaced token is labeled unacceptable, which implies that this replaced token should be rejected if it appears during the inference phase. For data construction, we save the pairs of hidden representations and labels $(h_z, label)$.

Our token label incorporates information about how closely the replaced token preserves the semantic information in the original response. We argue that `SelfJudge` is a more robust verification approach than the answer preservation-based approach, as our approach focuses on keeping the local semantics of the responses generated by the target model. Moreover, `SelfJudge` eliminates human bias when assessing the acceptable tokens.

**Discussion on the Semantic Preservation Score.** A straightforward approach to accepting the token from the draft model (accepting the replaced token $z$) is to compare the confidence of $y$ and $z$. Some prior works (Kim et al., 2023; Narasimhan et al., 2025) have compared the probabilities solely on the prefix context, i.e., $s_{\text{prefix}}$ in Eqn.(7), to accept high-confidence draft tokens.

Our bidirectional context approach enhances the criterion by incorporating the suffix from the original response, which reduces the uncertainty of replacement decisions. We formally define our semantic preservation score, $s(y, z_i)$, which can be decomposed to reveal its direct relationship with the prefix-only baseline. Through a Bayesian interpretation, our score can be expressed as:

$$
s(y, z_i) = s_{\text{prefix}}(y, z_i) + \underbrace{\log \left( \frac{P(y_{>i} \mid z_i, y_{<i})}{P(y_{>i} \mid y_i, y_{<i})} \right)}_{\text{Log Bayes Factor}}
\tag{8}
$$

This formulation demonstrates that our score begins with the baseline prefix score and refines it with an additional term, where the Log Bayes Factor quantifies the strength of evidence provided

by the suffix. If the Log Bayes Factor is large, it indicates that the suffix is more probable given the replacement token, thus increasing the score. Conversely, if it is small, the suffix provides evidence for the original token, penalizing the replacement score. This transforms the decision-making process from a simple probability comparison into a principled, evidence-based update. Since obtaining the future token probabilities during the inference phase is not possible, we use the learned verifier to predict the semantic preservation score online.

### 3.4 VERIFIER TRAINING

After collecting a dataset with semantically-preserved tokens, we can train a verifier that identifies acceptable tokens during the inference phase. We generate 69,432 token labels using 1,220 prompts in GSM8K train, LiveCodeBench and Dolly15k datasets. While the judge verifier could be any type of model, we primarily select simple linear models that utilize existing LLM hidden states as input features, following (Bachmann et al., 2025). Specifically, we employ basic logistic regression for recognizing semantic-preserved tokens in our approach. To prevent overfitting, we conduct a straightforward grid search across L2 regularization parameters.

## 4 EXPERIMENTS

### 4.1 EXPERIMENTAL SETTINGS

**Evaluation Protocol.** We comprehensively evaluate our proposed method by following the SD literature (Bachmann et al., 2025). We select Llama-3.1-8B/Llama-3.2-1B and Qwen-2.5-8B/Qwen-2.5-0.5B as the target and draft models, respectively. To measure inference efficiency, we report the average accepted length ($m$). However, relaxed verification methods can introduce the possibility of accepting incorrect tokens, which may lead to task performance degradation. Hence, we report task-specific performance metrics with the average accepted lengths. We conduct the experiments on five evaluation scenarios, including GSM8K (Cobbe et al., 2021), MATH-500 (Lightman et al., 2023), LiveCodeBench (Jain et al., 2025), CNN/DailyMail, and MMLU (Hendrycks et al., 2021). Note that we apply chain-of-thought prompting (Wang et al., 2025) while the original MMLU benchmark employs a multiple-choice prompt that encourages language models to directly answer among four choices. This modification allows us to simultaneously assess both QA task performance and inference efficiency, taking into account the trade-off between accuracy and the average accepted lengths. For more details of the experimental setup, refer to Appendix C.

**Baseline.** For baseline comparisons, we include standard SD (Leviathan et al., 2023) with rejection sampling, which preserves the target model's task performance. We also include top-k verification, which accepts draft tokens if they are among the target model's top-k tokens, as a naive relaxed verification approach. Both methods represent alignment-based verification approach, as they determine acceptance based on token-level matching with the target model's distribution. We also compare against judge verification approaches, specifically AutoJudge, which extends JudgeDecoding. We use AutoJudge instead of JudgeDecoding directly because the training data and implementation details of JudgeDecoding are not publicly available.

We train two versions of the AutoJudge verifier: AutoJudge and AutoJudge$_+$. Specifically, AutoJudge generates token labels based on the importance of each token to the final answer, using the training split of the GSM8K and LiveCodeBench datasets. While AutoJudge can assess token importance by checking whether a token replacement preserves the correctness on math and coding problems, this approach has limited applicability to open-ended NLP tasks. To address this limitation, we implement AutoJudge$_+$ by combining an LLMs-as-Judges framework (Li et al., 2024b) with AutoJudge. Specifically, we incorporate the instruction tuning dataset, Dolly15K, which encompasses diverse categories such as summarization, creative writing, and question answering. We train the verifier to act as a judge, asking it to determine if the original response and the token-substituted response are semantically equivalent.

**Implementation Details.** To set the threshold $\tau$ for semantic preservation labeling of SelfJudge, we use 100 sampled queries from GSM8K. In this math task, AutoJudge has identified ground-truth important tokens (i.e., tokens whose replacement changes the final answer). Specifically, we set $\tau = \mathsf{quantile}(\textit{semantic preservation scores of unacceptable tokens determined by AutoJudge}, 0.1)$ to ensure that all tokens rejected by AutoJudge are also rejected by SelfJudge. This conservative

Table 1: Performance comparison with SD baseline using Llama-3.1-8B/Llama-3.2-1B. $m$ denotes the average accepted length. Green = best performance gain compared to SD, Yellow = second place, Red = worst two. $\Delta_m$: Average change in accepted length from SD, $\Delta_{task}$: Average change in task performance from SD. More details on experimental settings can be seen in Appendix C.

| Method | GSM8K | MATH-500 | LiveCodeBench | MMLU | CNN/DM | Avg. Change |
| | $m$ / Acc. | $m$ / Acc. | $m$ / Pass@1 | $m$ / Acc. | $m$ / FC | $\Delta_m/\Delta_{task}$ |
|---|---|---|---|---|---|---|
| SD | 9.14 / 80.7 | 9.98 / 45.4 | 7.88 / 8.6 | 4.36 / 65.0 | 4.35 / 64.6 | - / - |
| Top-$k$ | 14.59 / 71.2 (-9.5) | 15.08 / 29.8 (-15.6) | 13.36 / 0.7 (-7.9) | 6.69 / 49.5 (-15.5) | 7.49 / 62.5 (-2.1) | +4.30 / -10.1 |
| AutoJudge-R | 9.71 / 80.1 (-0.6) | 11.23 / 41.0 (-4.4) | 15.78 / 0.7 (-7.9) | 4.47 / 64.6 (-0.4) | 4.36 / 64.5 (-0.1) | +1.96 / -2.7 |
| AutoJudge-F | 11.63 / 78.4 (-2.3) | 13.12 / 40.6 (-4.8) | 19.63 / 2.1 (-6.5) | 5.15 / 63.1 (-1.9) | 4.74 / 64.5 (-0.1) | +3.71 / -3.1 |
| SelfJudge-R | 10.09 / 80.7 (+0.0) | 10.91 / 43.0 (-2.4) | 8.12 / 9.7 (+1.1) | 5.14 / 64.4 (-0.6) | 4.92 / 65.5 (+0.8) | +0.69 / -0.2 |
| SelfJudge-F | 11.29 / 80.5 (-0.2) | 12.78 / 42.2 (-3.2) | 9.52 / 10.0 (+1.4) | 6.38 / 62.7 (-2.3) | 6.05 / 63.9 (-0.7) | +2.06 / -1.0 |

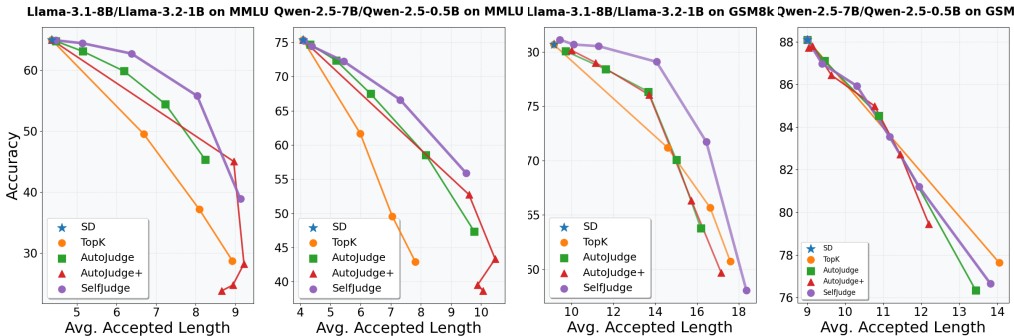

Figure 3: Speed/Performance comparison across different methods. We report the accuracy, with the corresponding average accepted length, by searching the threshold values of each method.

threshold guarantees high recall for answer-critical tokens of our method. Based on the established threshold $\tau$, SelfJudge generates token labels for full training data. The suffix length for computing semantic preservation score is set to 20. For reproducibility, we set the temperature of the target model to 0 in all experiments. We used NVIDIA V100 GPUs for all experiments.

## 4.2 PERFORMANCE COMPARISON ACROSS NLP TASKS

We first compare SelfJudge with baseline verification methods in a practical setting where we apply judge verification in diverse NLP scenarios. As judge verification methods have a trade-off between task performance and inference efficiency depending on the threshold value $\theta$, we set a fixed threshold for the judge verifier using a holdout set in advance. {Method}-R and {Method}-F denote applying $\theta$ that achieved the best recall and best F1 score, respectively. From Table 1, we make the following observations:

**Observartion1.** SelfJudge achieved significant speed-up over standard SD across various NLP tasks while demonstrating the smallest task performance degradation compared to SD. Particularly when we compare SelfJudge-F with AutoJudge-R, it shows much less performance degradation while achieving higher average accepted length across the GSM8K, MATH-500, MMLU, and CNN/DM datasets, resulting in greater inference speed improvements. Moreover, note that $\Delta_m$ of AutoJudge-R is inflated due to task accuracy dropping close to zero on LiveCodeBench. These suggest that our approach of accepting tokens based on semantic coherence outperforms AutoJudge's strategy of accepting tokens based on final answer correctness.

**Observation2.** SelfJudge has the advantage of consistently generalizing across all tasks. Specifically, existing baselines show significant performance loss on certain tasks. Top-$k$ ($k = 2$) verification achieves large speed-ups but substantially degrades task performance. AutoJudge-R partially achieves meaningful speed-up without significant performance drop compared to SD on GSM8K and MATH-500, but shows almost no speed-up on MMLU and CNN/DM. Moreover, it shows a severe decrease in Pass@1 on LiveCodeBench. In contrast, both SelfJudge-R and SelfJudge-F consistently achieve speed-ups across all datasets with minimal performance degradation among all compared methods.

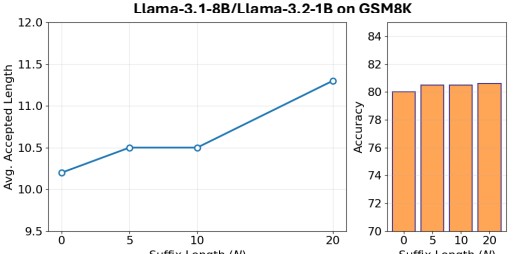 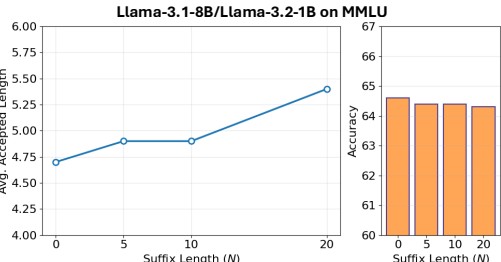

Figure 4: Performance of `SelfJudge` over a range of suffix length $N$. We compute the semantic score by including the likelihood computed on $N$ future tokens.

We argue that these results stem from the superiority of our semantic coherence-based token acceptance strategy. Specifically, `SelfJudge` is trained to accept tokens if they are semantically coherent in context, regardless of the task. In contrast, AutoJudge trains the verifier to accept tokens even when semantically different tokens appear in context, as long as the final answer remains identical to the original. This fundamental difference significantly impacts the robustness when the verifier encounters out-of-distribution tokens during actual inference scenarios.

### 4.3 Trade-off Analysis: Accepted Length vs. Task Performance

Since judge verification exhibits a trade-off between task performance and inference speed depending on the threshold setting, we performed additional analyzes using the Llama-3.1-8B / Llama-3.2-1B and Qwen-2.5-7B / Qwen-2.5-0.5B models. Specifically, we varied the threshold $\theta$ of each judge verifier. From Figure 3, we make two key observations. First, `SelfJudge` shows competitive performance to AutoJudge on MMLU with CoT prompting, indicating that the token generation criteria of `SelfJudge` correctly labels the token compared to answer-preservation-based methods AutoJudge and AutoJudge$_+$. These suggest that `SelfJudge` is consistently generalizing well in general reasoning tasks such as QA. Second, we observed that `SelfJudge` generally performs comparably to or better than AutoJudge methods on GSM8K. Although AutoJudge can determine the token label with the ground-truth answer for math/coding problems, our semantic-preserving approach remains competitive. This implies that determining token acceptance based on the semantic coherence in the local context is consistently better than judging token acceptance based on answer correctness, not only for general reasoning tasks but also for math and coding problems.

### 4.4 Semantic Preservation Score Analysis

When computing the semantic preservation score $s(y, z_i)$, we can adjust the length of the suffix $y_{>i}$. In Figure 4, we analyze the impact of the suffix length $N$ on performance. We observe that the highest accuracy is achieved when the suffix length $N = 20$, while $N = 0$ yields the worst performance. This implies that considering bidirectional context is beneficial for measuring token semantic preservation, resulting in improved judge decoding performance. Moreover, it indicates that suffix tokens provide stronger evidence for semantic preservation than prefix-only approaches, validating our approach.

### 4.5 Qualitative Analysis

We analyze the most frequently accepted token transformations by `SelfJudge` across three distinct domains: mathematical reasoning, coding, and general reasoning. First, across all task domains, we observed that `SelfJudge` consistently relaxes draft verification by accepting tokens, which change the format (", $\rightarrow$ .", "\n $\rightarrow$ \n\n") and equivalent sentence starters ("Let $\rightarrow$ To", "Next $\rightarrow$ Now"). Second, we observed the domain-specific transformations. For example, we find mathematical term substitutions like "solve $\rightarrow$ find" and "x $\rightarrow$ *", and coding style variations (":\n $\rightarrow$ :\n\n") reflecting different programming styles with identical functionality. These patterns confirm that `SelfJudge` achieved great inference efficiency by accepting tokens with minor token discrepancy. Please refer to Appendix D.3 for more examples.

## 5 CONCLUSION

In this paper, we introduce a paradigm shift for judge verification in SD, moving beyond the conventional reliance on human annotations and task-specific ground truths. We present `SelfJudge`, a novel framework that pioneers a self-supervised approach, leveraging the target model itself to assess semantic preservation. Our method automatically constructs a verifier training data by quantifying the contextual impact of token substitutions through a principled "semantic preservation score" derived from the model's own likelihoods. Empirical validation across a diverse suite of NLP benchmarks confirms that `SelfJudge` achieves substantial inference acceleration while maintaining high fidelity to the target model, establishing our method as a practical and scalable foundation for efficient large language model inference. Our current work adopts a simple linear verifier, and a broader exploration of more complex architectures remains an important next step. Consequently, we have not yet investigated the scaling laws that relate the volume of training data to verifier performance, an understanding of which will be crucial for improving the performance of `SelfJudge`.

ETHICS STATEMENT

This research contributes to making large language model inference more efficient and accessible, which benefits society by reducing computational costs of AI deployment. By enabling faster inference while maintaining output quality, our approach helps democratize access to advanced language models across different socio-economic regions. We acknowledge that improved AI efficiency may have broader societal implications and commit to transparent reporting of our methods to enable responsible adoption. In compliance with ICLR's code of ethics, we have carefully considered potential negative consequences and the limitations of our approach. While we introduce some relaxation in draft verification that could lead to semantic drift, our extensive evaluation across diverse tasks demonstrates minimal performance degradation compared to standard speculative decoding.

REPRODUCIBILITY STATEMENT

While our research did not directly involve human subjects as participants, we recognize that our findings may influence the development of AI systems that affect end users. We have conducted this research with rigorous methodology and report our findings accurately and honestly. All experimental results, including both positive and negative outcomes, are presented transparently. Our methods are described in sufficient detail to enable reproducibility, and we have made our implementation approach clear by reporting the experimental configurations.

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

Table 2: Data for training verifier of each judge decoding methods.

| Target Model | Method | Dataset | Num. of samples | Label Rate |
|---|---|---|---|---|
| Llama-3.1-8B | AutoJudge | GSM8K (train), LiveCodeBench (train) | 14896 | 13633 |
| | AutoJudge$_+$ | GSM8K (train), LiveCodeBench (train), Dolly15k | 39069 | 37261 |
| | SelfJudge | GSM8K (train), LiveCodeBench (train), Dolly15k | 69432 | 42253 |
| Qwen-2.5-7B | AutoJudge | GSM8K (train), LiveCodeBench (train) | 24748 | 13068 |
| | AutoJudge$_+$ | GSM8K (train), LiveCodeBench (train), Dolly15k | 55757 | 44067 |
| | SelfJudge | GSM8K (train), LiveCodeBench (train), Dolly15k | 95128 | 58482 |

## A  PROOF OF UNCERTAINTY REDUCTION USING BIDIRECTIONAL CONTEXT

In this section, we provide a formal, information-theoretic proof that incorporating bidirectional context strictly reduces the uncertainty of selecting acceptable tokens.

**Theorem 1.** *When the suffix* $\mathbf{x}_{>t}$ *provides additional information about a token* $X_t$ *given the prefix* $\mathbf{x}_{<t}$ *(i.e., the conditional mutual information* $I(X_t; \mathbf{x}_{>t} \mid \mathbf{x}_{<t}) > 0$*), the uncertainty about* $X_t$*, measured by conditional entropy, is strictly reduced when using the bidirectional context* $\mathbf{c}_t = (\mathbf{x}_{<t}, \mathbf{x}_{>t})$.

$$H(X_t \mid \mathbf{c}_t) < H(X_t \mid \mathbf{x}_{<t}) \tag{9}$$

**Proof.** The proof follows directly from the definition of conditional mutual information.

1. The definition of conditional mutual information between two random variables $A$ and $B$ given a third variable $C$ is:

$$I(A; B \mid C) = H(A \mid C) - H(A \mid B, C) \tag{10}$$

2. We substitute our variables: $A = X_t$, $B = \mathbf{x}_{>t}$, and $C = \mathbf{x}_{<t}$.

$$I(X_t; \mathbf{x}_{>t} \mid \mathbf{x}_{<t}) = H(X_t \mid \mathbf{x}_{<t}) - H(X_t \mid \mathbf{x}_{>t}, \mathbf{x}_{<t}) \tag{11}$$

3. By our definition, the bidirectional context is $\mathbf{c}_t = (\mathbf{x}_{<t}, \mathbf{x}_{>t})$. Therefore, the final term is equivalent to the entropy conditioned on the bidirectional context:

$$H(X_t \mid \mathbf{x}_{>t}, \mathbf{x}_{<t}) = H(X_t \mid \mathbf{c}_t) \tag{12}$$

4. The premise of our theorem states that $I(X_t; \mathbf{x}_{>t} \mid \mathbf{x}_{<t}) > 0$. Substituting this into the equation from step 2 gives:

$$H(X_t \mid \mathbf{x}_{<t}) - H(X_t \mid \mathbf{c}_t) > 0 \tag{13}$$

5. Rearranging the terms yields the final result:

$$H(X_t \mid \mathbf{c}_t) < H(X_t \mid \mathbf{x}_{<t}) \tag{14}$$

This concludes the proof, demonstrating that conditioning on bidirectional context logically leads to lower uncertainty.

### A.1  DISCUSSION ON THE PREMISE

It is crucial to understand the role of the premise, $I(X_t; \mathbf{x}_{>t} \mid \mathbf{x}_{<t}) > 0$. From a Natural Language Processing (NLP), this premise is not merely an assumption but a well-established empirical fact. Language is inherently structured, and for any non-trivial linguistic context, the suffix almost always contains information that helps resolve ambiguity or refine the prediction of a token (Devlin et al., 2019). Therefore, while our proof relies on a mathematical assumption, its real-world applicability is grounded in the fundamental nature of human language.

## B  VERIFIER TRAINING

### B.1  STATISTICS

Table 2 shows the training data of verifiers across judge decoding methods. Following previous works, AutoJudge uses math and coding datasets to train the judge verifier. It assigns a token

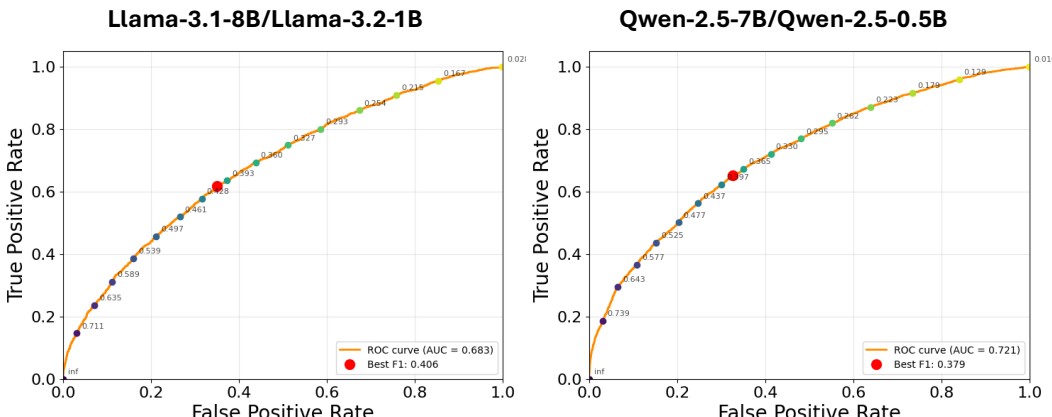

Figure 5: ROC curve of our judge verifier. We train each verifier for Llama-3.1-8B/Llama-3.2-1B and Qwen-2.5-7B/Qwen-2.5-0.5-B, individually. Red point represents threshold value with the best F1 score on validation set.

label as acceptable if the newly generated response after replacing a token preserves the original final answer in math and coding problems. To further utilize general datasets like Dolly15k, we implement AutoJudge$_+$ by generating token labels using LLMs-as-Judge approaches. Specifically, LLMs decide whether the original response and the alternative response are semantically [SAME] or [DIFFERENT]. We assign a token label as acceptable for tokens that results in [SAME].

## B.2 TRAINING LOGISTIC REGRESSOR

We employed a logistic regression classifier to address the binary classification problem, which determines acceptable and unacceptable token. We optimized hyperparameters through grid search over the $L_2$ regularization parameter $C$. We explored $C$ values from 0.001 to 100 across 10 logarithmically spaced points. Finally, we selected the verifier with the best ROC-AUC.

During inference phase, we apply the trained verifier our two-phase verification method. For the judge verification phase, we select the threshold $\theta$ to decide the acceptable tokens. We set the $\theta$ to value of the best recall and the best F1 as shown in Figure 5.

## B.3 DATASET GENERATION TIME

AutoJudge generates new responses by replacing each mismatched token and completing the text, which is computationally expensive. This requires generating responses *Num. of Queries* × *Num. of Mismatched Tokens* times. In contrast, our approach simply computes likelihood differences from the original response without generating new text. We only need to partially prefill a few tokens to calculate these likelihoods. As *partially prefilling* requires much less computational cost compared to *generation*, our approach is significantly efficient, making our method applicable to large models. Using Llama-3.1-8B on NVIDIA V100 GPUs, our `SelfJudge` method processed 69,432 data points in 8 hours, while AutoJudge processed only 14,896 data points in 82 hours.

## C EXPERIMENTAL CONFIGURATION

For mathematical reasoning evaluation, we utilize GSM8K (Cobbe et al., 2021) and MATH-500 (Lightman et al., 2023) using the standard train/test split and evaluation prompts in the benchmark. For code generation evaluation, we leverage LiveCodeBench (Jain et al., 2025). Since this benchmark does not provide predefined train/test splits, we implement a 4-fold cross-validation strategy. For the summarization task, we employ the CNN/DailyMail datasets (Nallapati et al., 2016) using the conventional test split. For general knowledge assessment, we evaluate on MMLU (Hendrycks et al., 2021), a multiple-choice QA benchmark. While the original MMLU

Table 3: Summarization Task Performance Comparison on Llama-3.1-8B/Llama-3.2-1B across various SD methods. $m$ is the averaged accepted length.

| Method | $m$ | ROUGE1 | ROUGE2 | ROUGE-L | FC |
|--------|------|--------|--------|---------|------|
| SD | 4.35 | 27.1 | 10.8 | 22.4 | 64.6 |
| TopK | 7.49 | 26.8 | 10.5 | 22.0 | 62.5 |
| AutoJudge-R | 4.36 | 27.0 | 10.8 | 22.4 | 64.5 |
| AutoJudge-F | 4.74 | 27.1 | 10.9 | 22.5 | 64.5 |
| SelfJudge-R | 4.92 | 27.5 | 11.0 | 22.8 | 65.5 |
| SelfJudge-F | 6.05 | 26.2 | 10.3 | 21.5 | 63.9 |

benchmark employs a multiple-choice prompt that encourages language models to directly answer among four choices, we apply chain-of-thought prompting (Wang et al., 2025). This modification allows us to simultaneously assess both QA task performance and inference efficiency, taking into account the trade-off between accuracy and the average accepted lengths.

## C.1 MATH REASONING BENCHMARKS

GSM8K (Cobbe et al., 2021) is a dataset of high-quality, diverse grade-school math word problems. We used 1,319 test problems for our evaluation. The MATH-500 dataset contains a subset of 500 problems from the MATH benchmark, which was created by (Lightman et al., 2023). For our evaluation, we used a prompt that required the answer format to appear between [answer] and [\answer]. After extracting the answers, we normalized the format by removing uppercase letters and certain operators. We finally check the correctness of the answer with ground truths to measure accuracy.

## C.2 CODING BENCHMARK

LiveCodeBench (Jain et al., 2025) is a continuously updating coding benchmark for evaluating Large Language Models. It evaluates LLMs across multiple capabilities including code generation, self-repair, test output prediction, and code execution. We used release version 0.0.5, which contains 880 programming problems, and applied 4-fold cross-validation to create training and test datasets. We evaluated performance using the Pass@1 metric, which measures the percentage of problems successfully solved on the first attempt when evaluated against comprehensive test suites.

## C.3 SUMMARIZATION BENCHMARK

The CNN/DailyMail dataset (Nallapati et al., 2016) is an English-language dataset containing over 300,000 unique news articles written by journalists at CNN and the Daily Mail. We evaluated summarization quality on 11,490 test samples from version 2.0.0, which supports both extractive and abstractive summarization. Since our evaluation focuses on relaxed speculative decoding methods that maintain token-level equivalence with standard decoding, traditional metrics like ROUGE cannot effectively differentiate between methods—all produce identical outputs in terms of content. Therefore, we instead report factual consistency scores to assess summarization quality (Laban et al., 2022). We measure factual consistency using a Natural Language Inference (NLI) model to classify the relationship between source articles and generated summaries. Specifically, we identify sentences that are either entailed by or contradict the source material. The factual consistency score (FC) is computed as:

$$\text{Factual Consistency} = \frac{1}{M} \sum_{i=1}^{M} (\text{Entailments}_i - \text{Contradictions}_i) \tag{15}$$

where $M$ is the number of summary sentences evaluated. We employed the `deberta-large-mnli` model for NLI classification due to its strong performance on factual verification tasks. In Table 3, we can observe the result of summarization task in more details. As we mentioned, ROUGE scores, which depend on token-level alignment, do not show distinctive difference across methods. Hence, we selected the FC score as our summarization task metric, as it reflect more semantic information by considering entailment of fact when evaluating results.

Table 4: Data generation time comparison on Llama-3.1-70B/8B.

| Method | 53K labels | 1M labels |
|--------|-----------|-----------|
| AutoJudge | 120 hours | $\sim$100 days |
| SelfJudge | 8.5 hours | $\sim$6 days |

Table 5: Performance comparison on Llama-3.1-70B/8B. $m$ denotes the average accepted length.

| Method | GSM8K $m$ / Acc | MMLU $m$ / Acc |
|--------|-----------------|----------------|
| SD | 9.532 / 93.0 | 5.668 / 80.6 |
| AutoJudge | OOT | OOT |
| SelfJudge-R ($\theta = 0.28$) | 10.622 / 93.8 | 6.262 / 81.4 |
| SelfJudge-F ($\theta = 0.38$) | 11.663 / 94.0 | 6.955 / 79.8 |
| SelfJudge ($\theta = 0.5$) | 12.528 / 93.8 | 7.576 / 77.2 |

## C.4 MULTIPLE CHOICES BENCHMARK

MMLU (Hendrycks et al., 2021) is a massive multitask benchmark containing 14,042 multiple-choice questions across 57 tasks. The benchmark spans diverse domains including humanities, social sciences, hard sciences, and other fields, covering subjects from elementary mathematics to US history, computer science, and law. While the original MMLU evaluation prompts models to directly select answers from 'A', 'B', 'C', or 'D', we adopted the chain-of-thought prompting approach from (Wang et al., 2025) to measure both inference efficiency and task performance simultaneously.

## D ADDITIONAL EXPERIMENTS

### D.1 SCALING TO LARGE MODEL.

**Data Generation Cost and Scalability.** To examine the scalability of `SelfJudge` to larger models, we conducted additional experiments with Llama-3.1-70B/8B as target and draft models, respectively. Following the same protocol as our main experiments, we sampled 500 instances from GSM8K (train), 220 from Live Code Bench (train), and 500 from Dolly15k. SelfJudge generated a total of 53,318 token labels for verifier training.

Table 4 presents the data generation time comparison between `SelfJudge` and AutoJudge. Using four H100 GPUs, `SelfJudge` requires only 8.5 hours to generate these labels, while AutoJudge required 120 hours for the same dataset—achieving a 14$\times$ speedup. It is attributed to the fact that our data generation only requires the prefill operation of the target model's response given a mismatched index, whereas AutoJudge needs to complete the response generation to obtain the final answer.

While our results demonstrate that using 53,318 token labels as training data is sufficient for effective verifier, scenarios requiring higher-capacity verifiers may necessitate larger datasets. `SelfJudge` would require approximately 6 days to generate 1 million token labels, which remains manageable, whereas AutoJudge would require approximately 100 days, making data generation prohibitively expensive.

**Scaling to a larger models using Llama-3.1-70B/8B.** We evaluated `SelfJudge` by measuring both the accepted length ($m$) and task performance. Due to the prohibitive data generation time, AutoJudge could not complete the data generation within the experimental timeframe and is marked as Out of Time (OOT). Table 5 presents the results on GSM8K and MMLU benchmarks. The result confirms that `SelfJudge` maintains accuracy while improving accepted length and inference speed compared to SD, even on the 70B target model. This demonstrates that our approach scales effectively to larger models.

Table 6: Wall-clock throughput comparison on Llama-3.1-8B/Llama-3.2-1B using single A100 GPU with 1-way tensor parallelism. SelfJudge-F achieves substantial speedup over both baseline methods while maintaining competitive accuracy.

| Dataset | Method | Tokens/sec | Acc (%) |
|---------|--------|------------|---------|
| GSM8K | SD | 111.21 | 80.7 |
| | AutoJudge-R | 118.14 | 80.1 |
| | SelfJudge-F | **137.37** | 80.5 |
| MMLU | SD | 61.13 | 64.6 |
| | AutoJudge-R | 62.67 | 64.5 |
| | SelfJudge-F | **89.45** | 63.9 |

## D.2 WALL CLOCK TIME EVALUATION

To demonstrate that `SelfJudge` achieves genuine wall-clock speedup beyond theoretical improvements, we conducted comprehensive throughput experiments by implementing our method in vLLM, the de facto standard framework for production-grade LLM serving. We measured end-to-end inference throughput (tokens per second) on NVIDIA A100 GPUs, which represents the most widely adopted hardware configuration in both research and industrial deployments.

**Experimental Setup.** All experiments were conducted with consistent hyperparameters to ensure fair comparison. Specifically, we set the draft length $\gamma$ to 20 for GSM8K and 15 for MMLU across all methods. For the experiment with Llama-3.1-8B/Llama-3.2-1B, we employed single-way tensor parallelism (1-TP) on a single A100 GPU. We use the 4-way tensor parallelism (4-TP) on 4 A100 GPUs for the experiment with Llama-3.1-70B/8B.

**Results on Llama-3.1-8B/Llama-3.2-1B.** Table 6 presents the throughput and accuracy results for the Llama-3.1-8B/Llama-3.2-1B model pair. The results demonstrate that `SelfJudge-F` achieves substantially higher throughput than both standard speculative decoding (SD) and AutoJudge-R across both benchmark tasks. On GSM8K, `SelfJudge-F` delivers 137.37 tokens/sec, representing a **23.5%** improvement over SD (111.21 tokens/sec) and a **16.3%** improvement over AutoJudge-R (118.14 tokens/sec), while maintaining comparable accuracy (80.5% vs. 80.7%). More remarkably, on MMLU, `SelfJudge-F` achieves 89.45 tokens/sec, a **46.3%** speedup over SD (61.13 tokens/sec) and a **42.7%** improvement over AutoJudge-R (62.67 tokens/sec), with only a marginal accuracy difference (63.9% vs. 64.6%). These results confirm that the accepted length improvements we observed in prior experiments directly translate to real wall-clock speedup in production-grade serving systems.

**Results on Llama-3.1-70B/8B.** Table 7 presents the results for this larger model configuration. `SelfJudge-R` achieves 59.57 tokens/sec on GSM8K and 43.30 tokens/sec on MMLU, representing **11.5%** and **10.5%** improvements over SD, respectively. SelfJudge-F further pushes the throughput to 65.40 tokens/sec on GSM8K and 48.09 tokens/sec on MMLU, achieving **22.4%** and **22.7%** speedups over SD. Importantly, these speedups are achieved while maintaining or even improving accuracy (e.g., 94.0% vs. 93.0% on GSM8K), confirming that SelfJudge scales effectively to larger models without sacrificing generation quality.

## D.3 QUALITATIVE ANALYSIS

Tables 8, 9, and 10 present the most frequently accepted tokens by `SelfJudge` while they are rejected by Standard SD across three distinct domains: mathematical reasoning (GSM8K), coding (LiveCodeBench), and general reasoning (MMLU). These results reveal several key insights about the flexible nature of our approach and domain-specific characteristics of semantically preserving token modifications that do not compromise response quality. Domain-Agnostic Strengths. Across all three tasks, `SelfJudge` demonstrates consistent acceptance of tokens that preserve semantic meaning and logical equivalence. Mathematical terms like "solve $\rightarrow$ find" and calculate $\rightarrow$ find" in GSM8K, along with synonymous replacements such as number $\rightarrow$ element" in coding tasks, highlight the model's ability to identify semantically equivalent tokens that would not alter the fundamental meaning of the response. Moreover, `SelfJudge` successfully allows minor formatting

Table 7: Wall-clock throughput comparison on Llama-3.1-70B/Llama-3.1-8B using 4 A100 GPUs with 4-way tensor parallelism. AutoJudge could not be evaluated due to prohibitive training costs (OOT: Out of Time). SelfJudge demonstrates strong scaling properties on large target models.

| Dataset | Method | Tokens/sec | Acc (%) |
|---|---|---|---|
| GSM8K | SD | 53.45 | 93.0 |
| | AutoJudge-R | OOT | OOT |
| | SelfJudge-R | 59.57 | 93.8 |
| | SelfJudge-F | **65.40** | **94.0** |
| MMLU | SD | 39.19 | 80.6 |
| | AutoJudge-R | OOT | OOT |
| | SelfJudge-R | 43.30 | **81.4** |
| | SelfJudge-F | **48.09** | 79.8 |

Table 8: Freuquently appeared token acceptance by `SelfJudge` on Math Problem (GSM8K)

| Category | Origin | Alter. | Count |
|---|---|---|---|
| Similar Semantic | solve | find | 22 |
| | calculate | find | 7 |
| | x | * | 4 |
| | find | calculate | 4 |
| | number | amount | 2 |
| | amount | charge | 2 |
| Punctuation/ Format | .\n\n | . | 9 |
| | . | , | 6 |
| | . | .\n\n | 6 |
| | : | :\n | 5 |
| | ( | + | 4 |
| | To | \n | 4 |
| | \n\n | So | 3 |
| | , | . | 3 |
| | per | .\n\n | 3 |
| | :\n\n | :\n | 2 |
| | ( | . | 2 |
| | ). | ) | 2 |
| | .\n\n | .\n | 2 |
| | () | ( | 2 |
| | () | the | 2 |
| | \n\n | Since | 2 |
| | () | :\n | 2 |
| | : | :\n\n | 2 |
| Preposition/ Article | in | per | 6 |
| | / | per | 2 |
| | /p | per | 2 |
| | per | of | 2 |
| | a | one | 2 |
| | , | and | 2 |
| Sentence Starter | Let | To | 5 |
| | 1 | We | 4 |
| | Next | Now | 3 |
| | The | Since | 2 |
| | let | we | 2 |

Table 9: Freuquently appeared token acceptance by `SelfJudge` on Coding Problem (LiveCodeBench)

| Category | Origin | Alter. | Count |
|---|---|---|---|
| Line Break/ Indentation Changes | \n | \n\n | 44 |
| | :\n | :\n\n | 21 |
| | (8 spaces)\n | (7 spaces) | 7 |
| | (7 spaces) | (8 spaces)\n | 5 |
| | ):\n\n | ):\n | 4 |
| | ))\n\n | ))\n | 4 |
| | of | \n | 4 |
| | \n | \n(8 spaces)\n | 3 |
| | to | \n | 3 |
| | \n\n | \n | 3 |
| | ]\n | ]\n\n | 2 |
| | ]\n | ] | 2 |
| | .\n | .\n\n | 2 |
| | for | \n | 2 |
| | )}\n | )}\n\n | 2 |
| | (). | ()\n\n | 2 |
| | in | \n | 2 |
| | \n | of | 2 |
| | ]\n\n | ]\n | 2 |
| | )]\n | )]\n\n | 2 |
| Punctuation/ Brackets Changes | ():\n | s | 6 |
| | : | :\n\n | 4 |
| | . | ˙. | 3 |
| | == | != | 3 |
| Synonyms/ Similar Meaning | code | Python | 5 |
| | # | total | 4 |
| | number | length | 3 |
| | number | element | 3 |
| | return | find | 2 |
| | max | maximum | 2 |
| | count | number | 2 |
| | number | count | 2 |

Table 10: Freuquently appeared token acceptance by `SelfJudge` on General Reasoning (MMLU)

| Category | Origin | Alter. | Count |
|---|---|---|---|
| Punctuation/ Format Changes | . | , | 7 |
| | , | . | 7 |
| | . | .\n\n | 6 |
| | .\n\n | . | 5 |
| | , | )( | 2 |
| | because | .\n\n | 2 |
| | \n\n | \n | 2 |
| | , | a | 2 |
| | ( | , | 2 |
| | , | .\n\n | 2 |
| | .\n\n | , | 2 |
| | This | \n | 2 |
| | \n\n | In | 2 |
| | ), | ) | 2 |
| | ; | , | 2 |
| | ,' | , | 2 |
| | )\n\n | )\n | 2 |
| Articles/ Determiners Changes | a | the | 5 |
| | the | a | 5 |
| | the | this | 3 |
| | the | its | 2 |
| | a | an | 2 |
| | the | an | 2 |
| | we | the | 2 |
| | its | the | 2 |
| | known | a | 2 |
| | a | every | 2 |
| | in | a | 2 |
| Mathematical/ Logic Terms Changes | validity | correct | 6 |
| | Statement | statement | 5 |
| | correctness | correct | 3 |
| | answer | correct | 3 |
| Verbs/ Predicates Changes | let | we | 7 |
| | satisfy | make | 3 |
| | must | is | 3 |
| | use | multiply | 2 |
| | is | does | 2 |
| | is | means | 2 |

discrepancies such as ", → ." and "\n → \n \n". This flexibility enables speculative decoding to achieve significant latency reduction while preserving the semantic integrity and quality of the original target model response.

**Mathematical Reasoning Characteristics.** In GSM8K, `SelfJudge` exhibits a particular pattern of accepting mathematical vocabulary variations that preserve computational meaning. The frequent acceptance of transformations like x → *" and find → calculate" indicates that `SelfJudge` correctly identifies when mathematical operations remain functionally equivalent. Additionally, the acceptance of sentence starters like "Let → To" and "Next → Now" demonstrates that these discourse markers do not fundamentally alter the mathematical reasoning process, allowing for flexible problem-solving exposition styles without compromising the logical flow.

**Coding Domain Specificity.** LiveCodeBench results reveal how `SelfJudge` achieves high acceptance rates while maintaining code functionality. `SelfJudge` correctly accepts functionally

equivalent code transformations, such as whitespace normalization ("\n → \n\n") and formatting changes that do not affect program execution. Notably, synonym acceptance for programming concepts ("max → maximum", "count → number") shows the model's ability to recognize coding-specific terminology variations that preserve semantics. The acceptance of structural changes in brackets and punctuation effectively improve the efficiency of judge verification while preserving the target model's intention.

**General Reasoning Adaptability.** Table 5 exhibits frequent acceptance of transformations involving punctuation and formatting variations, indicating that such changes do not alter the semantic content or factual accuracy of responses. Particularly noteworthy is the acceptance of articles/determiners changes ("a → the", "the → a") and logical term modifications ( "validity → correct"), which suggests SelfJudge can distinguish between semantically equivalent expressions that convey identical meaning. We argue that minor linguistic variations with these token changes do not compromise the informational content or reasoning quality of the original response.

### D.4 HYPERPARAMETER SENSITIVITY ANALYSIS

**Prediction Threshold Analysis.** To comprehensively analyze the relationship between inference efficiency and task performance across varying threshold values, we conduct a systematic investigation into the threshold sensitivity of each method with respect to parameter $\theta$. In Figure 3, our experimental results demonstrate that SelfJudge consistently achieves superior performance compared to all baseline methods across diverse thresholds. This robust performance advantage indicates that SelfJudge represents a fundamentally superior approach, delivering optimal trade-offs between accepted sequence lengths and task-specific performance metrics while maintaining stability across different threshold settings. It implies that our method's effectiveness is not contingent upon fine-tuned hyperparameter selection, thereby enhancing its practical applicability and robustness in real-world deployment scenarios.

