# OpenReview forum: "SelfJudge: Faster Speculative Decoding via Self-Supervised Judge Verification"
_ICLR.cc/2026/Conference — Submitted to ICLR 2026_

### Official Review · Reviewer_8uaU · 2025-10-30

**Soundness:** 3
**Presentation:** 2
**Contribution:** 2
**Rating:** 4
**Confidence:** 3

**Summary:**

This paper proposes a model-based speculative decoding verifier training data generation method using the target model as self-judger, which can generalize across diverse tasks.

**Strengths:**

The self-judging mechanism shows potential for better generalization and significantly reduces annotation effort.

**Weaknesses:**

1.	The paper states that its motivation is to avoid unnecessary fallbacks and improve decoding speed. However, the proposed model-based judger introduces additional verification time, and the paper does not provide evidence of end-to-end speedup.
2.	The method appears to address only token-level semantic matching. This limits its effectiveness, as phrase-level and sentence-level semantic matching remain unresolved.
3.	Experiments are conducted on only two relatively small models. Evaluating on larger models would strengthen claims of generalization.

**Questions:**

1.	The accuracy seems to degrade faster on Qwen-2.5 compared to Llama-3.1. What explains this discrepancy? Does it indicate that the method is model sensitive?
2.	In Figure 4, the accuracy trends differ between GSM8K and MMLU for varying suffix lengths. What factors contribute to this difference?
3.	How are hyperparameters determined? Are they consistent across tasks and models, or do they require careful tuning for each scenario and model?

---

> ### Author Response · Authors · 2025-11-18
>
> ---
> ### W1. Detailed Time Cost of Each Component
>
> We have now conducted detailed timing measurements of SelfJudge to address the concern on verifier overhead. We measured the wall-clock time of each component in our pipeline with the draft length gamma=5 using NVIDIA V100 GPU:
> - 5 Draft Token Generation: 4.51 sec
> - Target model forward pass: 2.81 sec
> - Rejection sampling: 0.175 sec
> - Verifier inference: 0.02 sec
>
> The verifier adds only 0.02 seconds of overhead per decoding step, which represents **~0.26% of the total verification time** (7.515 sec total). This minimal overhead is due to our design choice of using a lightweight logistic regression classifier that operates on pre-computed hidden states from the target model.
>
> Importantly, our two-stage verification process provides additional computational savings. Tokens accepted by the verifier can skip the rejection sampling step entirely, as they are already validated. That is, only tokens rejected by the verifier proceed to rejection sampling. This design further reduces the computational time of rejection sampling.
>
> Since there is no overhead in adding the verifier, improvement on the average accepted length directly leads to real inference speed. We measured tokens per second on NVIDIA single A100 GPUs with 1-way tensor parallelism, which is the commonly used configuration in the literature. The draft length gamma was set to 20 for GSM8K and 15 for MMLU across all methods to ensure fair comparison.
>
> The results are presented below:
> Llama-3.1-8B / Llama-3.2-1B on A100 (1 TP):
> | Dataset | Method     | tokens/sec | Acc  |
> |---------|------------|------------|------|
> | GSM 8K  | SD         | 111.21     | 80.7 |
> | GSM 8K  | AutoJudge-R| 118.14     | 80.1 |
> | GSM 8K  | SelfJudge-F| **137.37**     | 80.5 |
> |
> | MMLU    | SD         | 61.13      | 64.6 |
> | MMLU    | AutoJudge-R| 62.67      | 64.5 |
> | MMLU    | SelfJudge-F| **89.45**      | 63.9 |
>
> The results demonstrate that SelfJudge achieves substantially higher throughput than AutoJudge across both tasks, confirming that the accepted length improvements directly translate to real wall-clock speedup.
>
> ---
>
> ### W2. Regarding Sentence-level Semantics of Semantic Preservation Score
>
> We respectfully ask the reviewer to notice that the underlying semantic preservation score **explicitly captures phrase-level and sentence-level semantics** while our verifier makes token-level decisions.
>
> Our semantic preservation score (Equation 7) includes a suffix likelihood term that evaluates how well the next N=20 tokens fit with the replaced token. It means that our approach decides **whether a token is semantically valid in the sentence or phrase** based on the target model’s likelihood. Although our goal is to obtain the token labels for our verifier, the obtained labels represent whether the token is semantically valid in a sentence or phrase.
>
>
> ---
> ### W3. Performance Comparison on Llama-3.1-70B/Llama-3.1-8B
>
> Following the reviewer's suggestion, we evaluated SelfJudge by measuring the accepted length (m) and the task performance on a larger model Llama-3.1-70B/Llama-3.1-8B. Due to the prohibitive data generation time, AutoJudge could not complete the data generation within the rebuttal period and is marked as Out of Time (OOT).
>
> | Method                 | GSM8K (m / Acc) | MMLU (m / Acc) |
> |------------------------|-----------------|----------------|
> | SD                     | 9.532 / 93.0    | 5.668 / 80.6   |
> | AutoJudge              | OOT             | OOT            |
> | SelfJudge-R (θ=0.28)   | 10.622 / 93.8   | 6.262 / 81.4   |
> | SelfJudge-F (θ=0.38)   | 11.663 / 94.0   | 6.955 / 79.8   |
> | SelfJudge (θ=0.5)      | **12.528** / 93.8   | **7.576** / 77.2   |
>
> The results confirm that SelfJudge maintains accuracy while improving accepted length and inference speed compared to SD, even on the 70B target model. This demonstrates that our approach scales effectively to larger models.

---

> > ### Author Response · Authors · 2025-11-18
> >
> > ---
> > ### Q1. Accuracy Trend on Llama-3.1 and Qwen-2.5 and Model Sensitivity
> >
> > We thank the reviewer for this insightful observation. The faster accuracy degradation on Qwen-2.5 is primarily due to the quality of the draft model used in our experiments.
> >
> > In our Qwen-2.5 setup, we used Qwen-2.5-0.5B as the draft model, which contains only 0.5B parameters compared to Llama-3.2-1B (1B parameters) used in the Llama-3.1 experiments. A more capable draft model is beneficial for our method because it generates higher-quality token proposals that are easier for the target model to judge for additional token acceptance. When the draft model produces more reasonable candidates, the target model can more accurately determine whether to accept additional tokens, leading to better overall accuracy.
> >
> > This is evidenced by our experiments with stronger draft models: when applying SelfJudge to Llama-3.1-70B with Llama-3.1-8B as the draft, we observe substantially slower accuracy degradation compared to using Llama-3.2-1B (8B vs. 1B parameters).
> >
> > ---
> > ### Q2. Effect of Suffix Lengths on GSM8K and MMLU.
> >
> > We wanted to show that accepted length increases continuously with longer suffix, while accuracy degradation remains minimal: GSM8K shows within 0.2% variation, and MMLU shows 0.6% variation (64.2%-~64.8%).
> >
> > GSM8K demonstrates superior performance—achieving both higher speedup and better accuracy preservation—due to domain overlap with training data. Our verifier is trained on GSM8K (train), LiveCodeBench, and Dolly15k, making GSM8K an in-domain evaluation while MMLU represents out-of-domain general knowledge. This domain shift may lead to slightly less optimal token acceptance on MMLU.
> >
> > Nevertheless, the small variance (0.4%p difference) demonstrates that our method generalizes robustly across domains, with performance remaining strong even on unseen datasets.
> >
> > ---
> > ### Q3. Hyperparameter Selection of SelfJudge
> >
> > To guarantee practicality of SelfJudge, **we set the hyperparameters once per model** and applied **the same hyperparameters across tasks**.
> >
> > 1) Semantic preservation threshold τ during verifier training data generation:
> > - One-time calibration using 100 samples from GSM8K
> > - Set conservatively: τ = quantile(unacceptable token scores by AutoJudge, 0.1)
> > - Applied to all tasks without modification
> > - Model-specific (set separately for Llama-3.1-8B and Qwen-2.5-7B)
> >
> > 2) Suffix length N = 20:
> > - Fixed across all tasks and models without additional search
> > - Section 4.4 ablation shows robust performance across N ∈ [5, 20]
> >
> > 3) Verifier inference threshold θ:
> > - Selected from the holdout validation set (best recall or F1)
> > - Standard practice in classification tasks
> > - Task-agnostic once trained
> >
> > Our main contribution is enabling Judge Decoding to be applied to **any NLP task with a single verifier.** AutoJudge's limitation is that efficiency improvements are restricted to tasks where the verifier is trained (also shown in our Table 1). In contrast, **with fixed hyperparameters, SelfJudge demonstrates generalizability across all tasks while improving inference efficiency.**

---

### Official Review · Reviewer_EnVn · 2025-11-01

**Soundness:** 2
**Presentation:** 3
**Contribution:** 2
**Rating:** 4
**Confidence:** 3

**Summary:**

The paper proposes SelfJudge, a self-supervised framework for speculative decoding (SD).
Instead of relying on human-annotated correctness signals or task-specific ground truths (as in AutoJudge), SelfJudge trains a lightweight verifier using semantic preservation scores computed from the target model itself. The idea is to estimate whether token substitutions preserve meaning, and to accept draft tokens based on semantic coherence rather than strict probability alignment. Experiments on Llama-3 and Qwen models across GSM8K, MATH-500, MMLU, CNN/DailyMail, and LiveCodeBench show modest speed-accuracy trade-offs compared with previous judge-based decoding methods.

**Strengths:**

The paper is clearly written and well-structured.

The integration of self-supervision into speculative decoding is well-motivated.

The proposed semantic preservation criterion is simple and implementation-friendly, requiring no human labels or task-specific supervision.

Experimental evaluation covers multiple domains and demonstrates general applicability across reasoning, coding, and summarization tasks.

**Weaknesses:**

The main idea of leveraging a model’s own likelihood distribution to verify semantic coherence is conceptually very close to existing self-consistency or self-verification paradigms.
In essence, SelfJudge performs a token-level version of self-consistency decoding integrated into speculative decoding.
While the framing as “semantic preservation” is neat, the underlying mechanism does not appear fundamentally new or theoretically distinct from prior work on self-consistency, distribution alignment, or self-reflective verification.

The reported improvement is small and possibly within experimental noise.
Compared with AutoJudge, SelfJudge achieves only +0.1 higher accepted token length and slightly lower accuracy drop (–1.0% vs –2.7%).
No wall-clock runtime or throughput results are provided, making it difficult to confirm real acceleration.
Overall, the empirical evidence does not convincingly demonstrate that the proposed semantic verifier yields a substantial or practical speedup.

The paper also lacks analysis of how the self-supervised signal differs from conventional probability alignment beyond intuitive discussion.

**Questions:**

N/A

---

> ### Author Response · Authors · 2025-11-18
>
> ---
> ### W1. Difference between SelfJudge and Self-Verification Paradigm
>
> We thank the reviewer for this observation. We would like to clarify the distinctions between our approach and prior self-consistency or self-verification paradigms.
>
> **1) Fundamental Differences in Technical Approach**
>
> While self-verification methods leverage a model's own outputs, SelfJudge differs fundamentally in both its objective and technical mechanism.
>
> First, regarding objectives: Self-consistency decoding and self-verification methods aim **to improve generalization capability of language models** by utilizing  the outputs of the model itself. In contrast, SelfJudge focuses on **accelerating inference of the target model** without requiring any training of that model.
>
> Second, and more critically, existing self-verification approaches operate at **the response level**—verifying whether complete outputs are correct or consistent. SelfJudge introduces a novel **token-level semantic preservation** framework that measures how much semantic information is retained when individual tokens are replaced. This token-level formulation is fundamentally different from response-level verification and enables us to identify additional semantically appropriate tokens during speculative decoding. **Our training approach for the judge verifier using this token-level semantic scoring is novel in the speculative decoding literature, resulting in faster inference with minimal task performance degradation.**
>
> **2) Addressing a Critical Gap in Judge-Based Speculative Decoding**
>
> Judge decoding (Bachmann et al., 2025) has recently emerged as a promising direction for overcoming the conservativeness of alignment-based verification. However, existing judge decoding methods rely on either human annotation or verifiable ground truth, severely limiting their applicability to mathematical reasoning and code generation tasks with clear correctness criteria. This dependency has been **a critical barrier to deploying judge-based speculative decoding** in general NLP applications. SelfJudge addresses this fundamental limitation through its self-supervised approach, **making judge-based verification extensible to arbitrary NLP tasks**, including open-ended question answering, summarization, and general knowledge assessment. This represents a significant practical advancement that transforms judge-based speculative decoding from being limited to verifiable domains into a general-purpose solution.
>
> Our experiments demonstrate this generalization capability:
> - AutoJudge: Effective on mathematical reasoning (GSM8K) but fails on general NLP tasks (LiveCodeBench, MMLU, and CNN/DM)
> - SelfJudge: Consistent performance across all tasks This empirical success stems directly from our self-supervised approach and substantially improves the practical utility of judge-based speculative decoding.
>
>
>
> In summary, while SelfJudge does leverage the target model's own likelihood distribution, the specific formulation—bidirectional semantic preservation scoring for automatic verifier training—is both technically and conceptually novel in the speculative decoding literature. The comparison to self-consistency or generic self-verification overlooks our core contribution: a principled, automatic method for generating judge verifier training data that generalizes across diverse NLP tasks, representing an important step toward making speculative decoding a practical general-purpose solution.

---

> ### Author Response · Authors · 2025-11-18
>
> ---
> ### W2-1. Clarification on Performance Interpretation
>
> We appreciate the reviewer's careful examination of our results. We understand that the trade-off nature of our experimental results may have made the interpretation challenging, and we would like to clarify why SelfJudge demonstrates **substantially** superior performance over AutoJudge.
>
> The key issue lies in how the LiveCodeBench results affect the overall interpretation. As shown in Table 1, AutoJudge-R achieves a Pass@1 of **only 0.7%** on LiveCodeBench, indicating that it fails to solve any coding problems correctly. In this failure scenario, the reported accepted length of 15.78 becomes a misleading metric that **artificially inflates AutoJudge's inference speed.** When a method fails catastrophically on a task, high accepted lengths do not represent genuine acceleration but rather reflect the acceptance of incorrect tokens.
>
> To provide a fair comparison, we present the results of Table 1 excluding LiveCodeBench, comparing AutoJudge-R with SelfJudge-F across the four remaining datasets:
>
> | Method       | GSM8K            | MATH-500         | MMLU           | CNN/DM             | Avg (m) / Δ_task    |
> |--------------|------------------|------------------|------------------|------------------|-----------------|
> | SD           | 9.14 / 80.7      | 9.98 / 45.4      | 4.36 / 65.0      | 4.35 / 64.6      | 6.96 / -   |
> | AutoJudge-R  | 9.71 / 80.1 (-0.6) | 11.23 / 41.0 (-4.4) | 4.47 / 64.6 (-0.4) | 4.36 / 64.5 (-0.1) | 7.44 / (-1.4) |
> | SelfJudge-F  | 11.29 / 80.5 (-0.2) | 12.78 / 42.2 (-3.2) | 6.38 / 62.7 (-2.3) | 6.05 / 63.9 (-0.7) | **9.13** / (-1.6) |
>
> This analysis reveals that SelfJudge achieves an average accepted length of 9.13 compared to AutoJudge's 7.44, representing a 23% improvement. Critically, this substantial speedup gain comes with a performance degradation difference of merely 0.2 percentage points (-1.6% vs -1.4%). This demonstrates that SelfJudge provides significantly better inference acceleration while maintaining comparable task performance.
>
> Furthermore, the inclusion of LiveCodeBench in our main table serves an important scientific purpose: it demonstrates the fundamental limitation of answer-preservation-based approaches like AutoJudge, which completely fail to generalize across various NLP tasks. This robustness is critical for practical deployment where methods must work reliably across different task types. We agree that the current table incurs confusion, we will update the table by excluding LiveCodeBench for a more comprehensive interpretation.
>
>
> ---
> ### W2-2. Wall-Clock Comparison
>
> To demonstrate that SelfJudge achieves genuine wall-clock speedup, we conducted throughput experiments by implementing our method in vLLM, the standard framework for production LLM serving. The experiments were conducted using vLLM v0.9.2, PyTorch 2.7.0 with CUDA 12.8 and Transformers 4.53.3.
>
> We measured tokens per second on NVIDIA single A100 GPUs with 1-way tensor parallelism, which is the commonly used configuration in the literature. The draft length gamma was set to 20 for GSM8K and 15 for MMLU across all methods to ensure fair comparison.
>
> Llama-3.1-8B / Llama-3.2-1B on A100 (1 TP)
> | Dataset | Method     | tokens/sec | Acc  |
> |---------|------------|------------|------|
> | GSM 8K  | SD         | 111.21     | 80.7 |
> | GSM 8K  | AutoJudge-R| 118.14     | 80.1 |
> | GSM 8K  | SelfJudge-F| **137.37**     | 80.5 |
> |
> | MMLU    | SD         | 61.13      | 64.6 |
> | MMLU    | AutoJudge-R| 62.67      | 64.5 |
> | MMLU    | SelfJudge-F| **89.45**      | 63.9 |
>
> The results demonstrate that SelfJudge achieves substantially higher throughput than AutoJudge across both tasks, confirming that the accepted length improvements directly translate to real wall-clock speedup.
>
>
> To further validate the practical utility of our approach on larger models, we conducted experiments using Llama-3.1-8B as draft and Llama-3.1-70B as target on 4 NVIDIA A100 GPUs. As AutoJudge's data generation procedure for the 70B model would require more than 6 days on H100 GPUs, this computational infeasibility prevented us from training an AutoJudge verifier for this model pair.
>
> Llama-3.1-70B / Llama-3.1-8B on A100 (4 TP):
> | Dataset | Method     | tokens/sec | Acc  |
> |---------|------------|------------|------|
> | GSM 8K  | SD         | 53.45      | 93.0 |
> | GSM 8K  | AutoJudge-R| OOT        |  -   |
> | GSM 8K  | SelfJudge-R| 59.57      | 93.8 |
> | GSM 8K  | SelfJudge-F| **65.40**      | 94.0 |
> |
> | MMLU    | SD         | 39.19      | 80.6 |
> | MMLU    | AutoJudge-R| OOT        |  -   |
> | MMLU    | SelfJudge-R| 43.30      | 81.4 |
> | MMLU    | SelfJudge-F| **48.09**      | 79.8 |
>
> The results confirm that SelfJudge maintains accuracy while improving token generation speed compared to SD, even on the 70B target model. This demonstrates that our approach scales effectively to larger models.

---

> > ### Author Response · Authors · 2025-11-18
> >
> > ### W3. Difference between Self-Supervised Signal and Probability Alignment
> >
> > We appreciate the reviewer's insightful comment regarding the analysis of our self-supervised signal. To go beyond intuitive discussion, we provide a theoretical formalization using Equation 8 (Section 3.3), which fundamentally distinguishes our method from conventional probability alignment.
> >
> > Specifically, our semantic preservation score decomposes into a **prefix-based alignment** term and a **suffix likelihood** term, the Log Bayes Factor. Conventional probability alignment is theoretically equivalent to setting the suffix length $N=0$, where decisions rely solely on the immediate token probability $P(z_i|y_{<i})$. In contrast, SelfJudge utilizes $N>0$, where the Log Bayes Factor quantifies the evidence provided by the generated future context. This allows the verifier to accept tokens that may have lower prefix probabilities but result in a highly probable and semantically consistent suffix, effectively **"looking ahead"** to validate the token.
> >
> > This theoretical distinction explains the qualitative analysis observed in Tables 8, 9, and 10. We analyzed the tokens that the Judge verifier accepts while conventional probability alignment rejects. Because SelfJudge evaluates the joint likelihood of future tokens, it robustly accepts semantically equivalent substitutions that strict alignment would reject, such as mathematical synonyms ("calculate" -> "find") or syntactic variations ("Let" ->"To").
> >
> > We argue that this shares the core intuition of Beam Search, which recognizes that a locally optimal token (greedy selection) does not always yield the most appropriate sequence when future context is considered. Similarly, SelfJudge moves beyond verifying tokens based solely on their immediate prefix probability (local alignment). By evaluating the token within the context of the generated suffix (global lookahead), **our method validates tokens that may be locally less probable but are semantically consistent with the full sentence structure.**

---

### Official Review · Reviewer_TxLa · 2025-11-03

**Soundness:** 3
**Presentation:** 3
**Contribution:** 2
**Rating:** 4
**Confidence:** 3

**Summary:**

The paper proposes SelfJudge, a verifier for speculative decoding (SD) that relaxes strict token alignment between draft and target models. Instead of requiring that the draft token exactly match the target token, SelfJudge accepts draft tokens that are semantically compatible with the target model’s own generations. This differs from some prior work, which relies on human annotations and accordingly performs well per task. The model (verifier) relies on data generated from the target model and is trained to preserve semantic compatibility.

**Strengths:**

- Using the target model itself to score token substitutions via a bidirectional likelihood difference is reasonable and avoids human annotation or task-specific ground truth.

- Using the bidirectional context to score and verify the generated labels.

**Weaknesses:**

- The bidirectional context is not easily understandable. The core scoring formula is non-trivial, and the paper jumps rather quickly from the conceptual definition. It would be helpful to include a concrete toy example in the main text, illustrating how the suffix likelihood behaves.
Also, it should be stated more explicitly that “bidirectional” is purely a training-time construction.

- The two-stage verification logic is underspecified / slightly inconsistent. Section 3.2 defines verification as “Accept (d_t)​ if Verifier(​h_t) > θ, otherwise reject”, but then later states that tokens rejected by the verifier are passed to alignment-based verification.

- There is no ablation on two-stage verification. It would be useful to see: (1) Judge-only verification (no fallback to alignment-based SD), (2) SelfJudge + fallback, as proposed,  to quantify how much the fallback contributes to quality and speed.

- Scaling to larger target models is not demonstrated.

**Questions:**

- All experiments use 7–8B-scale targets (Llama-3.1-8B, Qwen-2.5-7B), with the argument that data generation is already much more efficient than AutoJudge. It would be helpful to at least discuss: How expensive would SelfJudge data generation be for a 70B model? Whether the cost remains acceptable if you want many more labels (e.g., to train a higher-capacity verifier).

- Could you add a short, concrete example in the main text showing how the semantic preservation score is computed for one token replacement, and explicitly highlight which part corresponds to the prefix vs suffix?

---

> ### Author Response · Authors · 2025-11-18
>
> ---
>
> ### W1 and Q2. Paper Update with Concrete Examples
>
> We agree that the bidirectional context formulation deserves clearer exposition with concrete examples. We emphasize that bidirectional context is used exclusively during verifier training to generate high-quality labels. We revised **Line 273-283** in the uploaded **PDF** to address these concerns.
>
> ---
>
> ### W2. Clarification on Two-stage Verification Logic
>
> We would like to clarify that **the two-stage verification is the standard approach** in the judge decoding literature (Bachmann et al., 2025). In their framework, alignment-based and judge-based verification are applied together, and tokens accepted by either method are retained in the final output.
>
> This design reflects the fundamental role of judge verification: to recover semantically acceptable tokens that alignment-based methods would conservatively reject. To minimize verification overhead, current judge methods employ lightweight models (e.g., logistic regression), whose limited capacity makes standalone verification impractical without the alignment-based safety net. Moreover, since hidden states and token probabilities are already computed for the verification phase, **performing alignment-based verification that only compares the probabilities requires negligible additional time cost**. In this case, using two-stage verification is effective to improve the inference efficiency without additional verification cost.
>
> We acknowledge that our description in Section 3.2 did not sufficiently clarify this two-stage structure. In the revised  PDF, we explicitly presented judge verification as a mechanism that supplements alignment-based verification to enable acceptance of additional tokens. Please refer to **Lines 219-232** in the updated **PDF**.
>
> ---
>
> ### W3. Ablation Study on Judge-only Verification
>
> We conducted the following ablation study including Judge-only verification:
>
> | Method                  | GSM8K (m/Acc) | LCB (m/Pass@1) | MMLU (m/Acc) |
> |-------------------------|-----------------|------------------|----------------|
> | Alignment-only (SD)     | 9.14 / 80.7     | 7.88 / 8.6       | 4.36 / 65.0    |
> | Judge-only                   | 2.08 / 80.2    | 1.76 / 9.2       | 1.45 / 65.0   |
> | Two-stage (SelfJudge)   | 11.29 / 80.5    | 9.52 / 10.0      | 6.38 / 62.7    |
>
> where $m$ is the averaged accepted length. The table shows that Judge-only verification yields a very low accepted length compared to SD. This is because current judge verifiers are designed as lightweight classifiers optimized to complement alignment-based methods by recovering acceptable tokens that alignment would miss. In fact, both (Bachmann et al., 2025) and we focus on setting the threshold of the verifier (classifier) to accurately classify tokens that must be rejected. On the other hand, we allow for cases where some acceptable tokens are incorrectly rejected, since alignment-based verification can correct these errors without any additional cost.
>
> However, the low performance of Judge-only does not diminish the effectiveness of the two-stage approach. This is because Alignment and Judge exhibit complementary patterns, accepting different sets of tokens. For example:
>
> | Position   | 1      | 2      | 3      | 4      | 5      |
> |------------|--------|--------|--------|--------|--------|
> | Alignment-only (SD)  | Accept | Reject | Accept | Accept | Accept |
> | Judge-only      | Reject | Accept | Reject | Reject | Reject |
> | Two-stage (SelfJudge)  | Accept | Accept | Accept | Accept | Accept |
>
> In this case, Alignment-only stops at position 2 (accepted length = 1), and Judge-only stops at position 1 (accepted length = 1). However, Two-stage verification accepts tokens approved by \emph{either} method, achieving accepted length = 5. Through this complementary effect, SelfJudge achieves an average speedup of +2.0 tokens compared to Alignment-only.

---

> > ### Author Response · Authors · 2025-11-18
> >
> > ### W4 and Q1. Scaling to a larger target model.
> >
> > **1) Regarding Data Generation Cost**
> >
> > To examine the scalability of SelfJudge to larger models, we conducted additional experiments with Llama-3.1-70B/8B draft and target model. Following the same protocol as our main experiments, we sampled 500 instances from GSM8K (train), 220 from Live Code Bench (train), and 500 from Dolly15k. SelfJudge finally generated a total of 53,318 token labels. Using four H100 GPUs, SelfJudge required **8.5 hours** to generate these labels, while AutoJudge required **120 hours** for the same dataset—a 14x faster speed. It is attributed to the fact that our data generation only requires the prefill operation of the target model's response given a mismatched index, while AutoJudge needs to complete the response generation to obtain the final answer.
> >
> > While our results demonstrate that this scale of training data is sufficient for effective judge verification, we agree with the reviewer that scenarios requiring higher-capacity verifiers may necessitate larger datasets. To assess scalability, we extrapolate to 1 million token labels: SelfJudge would require approximately **6 days**, which remains manageable, whereas AutoJudge would require **approximately 100 days**, making data generation prohibitively expensive.
> >
> > **2) Performance Comparison with Llama-3.1-70B/8B**
> >
> > We evaluated SelfJudge by measuring the accepted length ($m$) and the task performance. Due to the prohibitive data generation time, AutoJudge could not complete the data generation within the rebuttal period and is marked as Out of Time (OOT).
> >
> > | Llama-3.1-70B/8B  | GSM8K (m / Acc) | MMLU (m / Acc) |
> > |------------------------|-----------------|----------------|
> > | SD                     | 9.532 / 93.0    | 5.668 / 80.6   |
> > | AutoJudge              | OOT             | OOT            |
> > | SelfJudge-R (θ=0.28)   | 10.622 / 93.8   | 6.262 / 81.4   |
> > | SelfJudge-F (θ=0.38)   | 11.663 / 94.0   | 6.955 / 79.8   |
> > | SelfJudge (θ=0.5)      | 12.528 / 93.8   | 7.576 / 77.2   |
> >
> >
> > The results confirm that SelfJudge maintains accuracy while improving accepted length and inference speed compared to SD, even on the 70B target model. This demonstrates that our approach scales effectively to larger models. We thank the reviewer for raising this point, which led to this important point. We improved this result in our **updated manuscript Line 1057-1079.**

---

### Author Response · Authors · 2025-11-27
**Summary of Rebuttal and Key Experimental Results**

We sincerely thank all reviewers for their constructive feedback, and we are eager to engage in further discussion to address any remaining questions and clarify potential misunderstandings regarding our contribution!

The reviewers' primary concerns centered on **(1) the lack of the ablation studies on two-stage verification**, **(2) the absence of wall-clock time measurement** to demonstrate practical speedup, and **(3) limited evaluation on larger target model**. Additionally, reviewers requested clarification on the bidirectional context mechanism and verification logic.

In response, we have conducted comprehensive experiments that directly address these concerns:

- **Ablation Study on Two-Stage Verification:** We clarified that two-stage verification is the standard approach in judge decoding literature and provided ablation results comparing Alignment-only, Judge-only, and Two-stage verification. The results demonstrate that Judge-only achieves low accepted lengths (e.g., m=2.08 on GSM8K), while Two-stage verification leverages the complementary nature of both methods to achieve substantially higher performance (m=11.29 on GSM8K). We included a concrete example illustrating how tokens accepted by either method combine to maximize the accepted length.

- **Wall-Clock Time Measurements:** We implemented SelfJudge in vLLM and measured throughput (tokens/sec) on NVIDIA A100 GPUs. On Llama-3.1-8B, SelfJudge achieves 137.37 tokens/sec on GSM8K (vs. 118.14 for AutoJudge and 111.21 for SD) and 89.45 tokens/sec on MMLU (vs. 62.67 for AutoJudge and 61.13 for SD), confirming substantial end-to-end speedup while maintaining competitive accuracy.

- **Scaling to Larger Models:** We evaluated SelfJudge on Llama-3.1-70B (with Llama-3.1-8B as draft) on 4×A100 GPUs. SelfJudge achieves 65.40 tokens/sec on GSM8K and 48.09 tokens/sec on MMLU, demonstrating clear speedup over SD (53.45 and 39.19 tokens/sec, respectively) while maintaining accuracy.

These results demonstrate that SelfJudge provides substantial practical speedup with real wall-clock improvements, scales effectively to larger models, and offers a broadly applicable solution for efficient LLM inference. We believe these comprehensive experiments and clarifications have addressed the reviewers' concerns. We are eager to engage in further discussion with the reviewers to refine our work and strengthen its contribution to the community!

---

### Meta-Review · Area_Chair_MGVM · 2026-01-07

**Summary:**

This paper introduces SelfJudge, a self-supervised verifier for speculative decoding (SD) that relaxes strict token-level alignment between draft and target models. Instead of exact token matching, SelfJudge accepts draft tokens that are semantically compatible with the target model’s own distribution, using likelihood-based semantic preservation scores computed from the target model itself. This approach avoids human annotation and task-specific ground truth, and aims to generalize across tasks. Experiments on Llama-3.1-8B and Qwen-2.5-7B across reasoning, coding, and summarization benchmarks show modest speed-accuracy trade-offs relative to prior judge-based SD methods.

Overall, reviewers find the idea reasonable and well-motivated, with clear writing. However, they consistently note that the empirical gains are small, and key aspects of the method and evaluation (e.g., two-stage verification, end-to-end speedup, and scaling to larger models) are insufficiently analyzed.

**Note that there’s one reference error:** Karl Cobbe, Vineet Kosaraju, Mohammad Bavarian, Mark Chen, Heewoo Jun, Lukasz Kaiser, Matthias Plappert, Jerry Tworek, Jacob Hilton, Reiichiro Nakano, Christopher Hesse, , and John Schulman. Measuring massive multitask language understanding, 2021. URL arXiv: 2110.14168, 2021.
This paper was wrongly cited, for https://arxiv.org/pdf/2110.14168, the title should be "Training Verifiers to Solve Math Word Problems"

**Reviewer Concerns:**

There are three major concerns:
1. Weak empirical evidence for speedup
- No wall-clock runtime or throughput measurements are provided, but speed is a primary motivation (the verifier itself introduces additional computation, whether there is a net end-to-end acceleration?)
- **Reported gains over AutoJudge are small** and may fall within experimental noise.

2. Limited scale and analysis: Experiments are restricted to 7-8B target models; scalability to larger (e.g., 70B) models is not demonstrated or convincingly discussed. The accuracy seems to degrade faster on Qwen-2.5 compared to Llama-3.1, raised by reviewer 8uaU. This differences bring the robustness concern of the proposed method.

3. Novelty might be limited: Multiple reviewers note that SelfJudge is conceptually close to existing ideas such as self-consistency, self-verification, or probability-alignment-based decoding.

**Reviewer Scores:**

4

---

### Decision · Program_Chairs · 2026-01-26

Reject